# Analysis of Culturable Bacterial Diversity of Pangong Tso Lake via a 16S rRNA Tag Sequencing Approach

**DOI:** 10.3390/microorganisms12020397

**Published:** 2024-02-17

**Authors:** Pooja Yadav, Joyasree Das, Shiva S. Sundharam, Srinivasan Krishnamurthi

**Affiliations:** 1Microbial Type Culture Collection & Gene Bank (MTCC), CSIR-Institute of Microbial Technology, Sec-39A, Chandigarh 160036, India; smile.pooja92@gmail.com (P.Y.); joyasree797@imtech.res.in (J.D.); shivas@imtech.res.in (S.S.S.); 2Academy of Scientific and Innovative Research (AcSIR), CSIR-HRDC Campus, Ghaziabad 201002, India

**Keywords:** Pangong Tso lake, amplicon sequencing, CFU, 16S rRNA

## Abstract

The Pangong Tso lake is a high-altitude freshwater habitat wherein the resident microbes experience unique selective pressures, i.e., high radiation, low nutrient content, desiccation, and temperature extremes. Our study attempts to analyze the diversity of culturable bacteria by applying a high-throughput amplicon sequencing approach based on long read technology to determine the spectrum of bacterial diversity supported by axenic media. The phyla Pseudomonadota, Bacteriodetes, and Actinomycetota were retrieved as the predominant taxa in both water and sediment samples. The genera *Hydrogenophaga* and *Rheinheimera*, *Pseudomonas*, *Loktanella*, *Marinomonas*, and *Flavobacterium* were abundantly present in the sediment and water samples, respectively. Low nutrient conditions supported the growth of taxa within the phyla Bacteriodetes, Actinomycetota, and Cyanobacteria and were biased towards the selection of *Pseudomonas*, *Hydrogenophaga*, *Bacillus*, and *Enterococcus* spp. Our study recommends that media formulations can be finalized after analyzing culturable diversity through a high-throughput sequencing effort to retrieve maximum species diversity targeting novel/relevant taxa.

## 1. Introduction

Freshwater lakes are an ideal spot for understanding the critical role of the microbiome in ecosystem shaping [1]. Altitude and salinity are some of the major deterministic factors affecting patterns of microbial community of a lake ecosystem [2,3,4]. High-altitude lakes are comparatively pristine with a unique mix of microbial diversity selected by severe environmental pressures (i.e., radiation, temperatures, nutrient fluctuations, etc.) that reduce species richness and increase prokaryotic dominance [5,6,7,8,9]. Recently, psychrophilic and psychrotolerant microorganisms of cold habitats have drawn tremendous attention due to the complex secondary metabolisms assisting their survival, adaptation, and evolution. Perhaps these cold habitats could be an asset of new taxa, genes, and metabolites with remarkable biotechnological applications [10,11,12]. Thus, a detailed study is required to gain information about the relationship between lacustrine ecosystem function and their resident microbial diversity.

The Himalayan range lakes are one of the most fragile and unique ecosystems, with 4699 lakes being documented in the region according to an ISRO study [13]. Pangong Tso, the largest high altitude brackish water lake, possesses multiple freeze–thaw cycles, nutrient scarcity, salinity, and high UV radiation, which makes it the most stressed ecosystem in the Indian Himalayan region. Microbial diversity analysis of the lake has been previously undertaken using both culturable and molecular approaches. Yadav et al. [14] described the composition of psychrotrophic bacteria by using traditional cultivation-based approaches and identified the phylum Bacillota (genus *Bacillus*) to be predominant with other reports targeting psychrotrophic bacteria in an array of biotechnological applications and their role in biogeochemical cycles [15,16] Culture-independent analysis has identified Pseudomonadota (genera *Loktanella*, *Rhizobium*, *Marinobacter*, *Pseudomonas*, *Methylophaga*) as the dominant taxa followed by Bacteroidota (genus *Gramella*) and *Bacillota* (genus *Zunogwangia*) [16,17,18] and an analysis of a soil–water mix collected beneath frozen ice revealed the dominance of Bacteroidota followed by Bacillota and Pseudomonadota [19]. Further, in high-altitude lakes (Pangong Tso, Dashair, Gurudongmar, Chandratal, Dal) of the Indian trans-Himalayan region [20] and in other continents (Spain, Dry-Andes, Australian Alps, and California [21,22,23,24]), similar patterns of limited bacterial diversity have been reported. However, despite these insights, routine culturable strategies are still important for the isolation, taxonomic, and metabolic characterization of pure strains as they decipher the key roles in biogeochemical cycles, functional activities in plant growth promotion, and cold active enzymes. One of the biggest limitations of studying structural and functional microbial diversity is the inability to cultivate the majority of bacteria [25]. To address these knowledge gaps, we have combined high-throughput next-generation sequencing based on long read technology with conventional cultivation techniques to understand the bacterial diversity of the lake ecosystem and how media formulation affects the scale of retrieval.

## 2. Materials and Methods

### 2.1. Sample Collection

The Pangong Tso lake is located in the Eastern part of Leh, Ladakh (33°57.560′ N and 078°25.627′ E), at a height of about 4350 m A.S.L. (above mean sea level). It extends from India to China, with only 40% of its length in India. Samples (both sediment and water) were collected in the month of September 2017 in 50 mL sterile falcon tubes (Tarsons, Appendix A) and whirl pack bags (1 L, Hi-Media, Thane, MH, India). Samples were transported to the laboratory within 48 h after collection. No permit was required for sample collection and it was performed with support from the local DRDO laboratory.

### 2.2. Physico-Chemical Parameters

The pH of the samples (water and sediment) was measured on-site using pH strips [Hi-Media], which were further analyzed in the laboratory using a pH meter [(EUTECH instruments pH 700)]. The temperatures of the water and sediment samples were recorded using a thermometer during sample collection. TDS, SO_4_^2−^, ammoniacal nitrogen, NO_3_, NO_2_, and Cl were analyzed as per standard protocols [26]. The analyses of Ca, Mg, Zn, Mn, Fe, B, Cu, Co, Mo, Cd, Cr, Al, B, Ni, Pb, Hg, Li, Va, Se, Si, and As were followed as per AOAC 990.08 standards [27]. TOC, TKN, conductivity, and salinity were recorded as per IS 2720 [28], AOAC 955.04 [29], APHA 2510B [30], and APHA 2520B [31] standards.

### 2.3. Sample Processing and Community DNA Isolation

About 1 g of sediment sample (PSD) was diluted in 10 mL of autoclaved distilled water and subjected to serial dilution by a factor of 10^−1^ to 10^−7^. Water samples (PW) were filtered through a 0.22 µm filter (Merck Millipore Ltd., Darmstadt, Germany) and the filter was suspended in 10 mL autoclaved distilled water in 50 mL falcon tubes (Tarsons, Kolkata, India), vortexed briefly, and serial dilutions were prepared (10^−1^ to 10^−7^). About 0.1 mL of the sediment and water dilutions was plated in duplicates on three types of media formulation prepared at two different pH levels (8.0 and 9.0), i.e., Reasoner’s 2A broth (R2A; Hi-Media), R2A broth, and tryptone soya broth (TSB, Hi-Media) diluted 100 times (R2A100) and 500 times (TSBA500) (Appendix A), respectively, with distilled water, solidified with 1.5% (*w*/*v*) extra-pure agar (Hi-Media). The plates were incubated at 25 °C for up to 8 weeks considering slow growing bacteria, and the colony forming units (CFUs) were recorded. After incubation, colonies were flooded with sterile distilled water (2–5 mL) and bacterial suspensions were pooled together from all the dilutions (10^−1^ to 10^−7^) and pH (8.0–9.0), for all the media (Figure 1).

Each pooled sample (PW_R2A, PW_R2A100, PW_TSBA500, PSD_R2A, PSD_R2A100 and PSD_TSBA500) and un-plated sediment (PSD)–water samples (PW), hereafter referred to as media and direct samples, respectively, were processed for DNA isolation using a FastDNA spin kit according to the manufacturer’s recommendations (MP Biomedical India Pvt. Ltd., Navi Mumbai, India).

### 2.4. Sequencing and Data Analysis

The total community DNA from all samples was subjected to 16S rRNA gene amplification with forward primer (5′-AGAGTTTGATCMTGGCTCAG-3′) and reverse primer (5′-CGGTTACCTTGTTACGACTT-3′) targeting the full-length gene (1.5 Kbp) amplification covering regions V1-V9 with LongAmp Taq 2× master mix (NEB). The thermocycling procedure involved an initial denaturation step at 95 °C for 1 min (1 cycle) followed by 25 cycles at 95 °C for 20 s, 25 cycles at 55 °C for 30 s (annealing), 25 cycles at 65 °C for 2 min (extension), and a final extension at 65 °C for 5 min (1 cycle). The PCR products were purified by using 1× Ampure XP beads (Beckmann Coulter, Brea, CA, USA). The purified PCR amplicons from each sample were pooled at an equimolar concentration. The pooled barcoded samples were then subjected to sequencing adapter ligation using the 16S Barcoding Kit (SQK-RAB204). Sequencing was performed on MinION Mk1b (Oxford Nanopore Technologies, Oxford, UK) using SpotON flow cell (FLO-MIN107) in a 48 h sequencing protocol on MinKNOW 1.10.11 with an error rate of 6–8%. Albacore v2.3.4 was used for basecalling with default parameters, which filters the reads having a Q score more than 7 that were considered as pass data and demultiplexed. The trimming of adapter sequences and chimeric sequences of raw data was performed using the Porechop tool and fastp reads were converted to fasta format using the seqtk tool. The reads were taxonomically assigned using Kraken2 (Galaxy Version 2.1.1) [32] using the RDP database (2020) with the default parameters. Data visualization was performed with the Pavian metagenomic tool [33]. In order to compare the bacterial diversity indices, PAST software version 3.1 [34] was used for the statistical analysis. The Shannon, Simpson, chao-1 diversity indices, and test of significant differences were performed using the Euclidean square root of the sum of the square distance clustering method. Principle component analysis (PCA) for abundant data normalization at the genus and species level was displayed by using the clustvis web tool [35]. For clustering analysis, the taxa with less than <0.1% were excluded from further analysis. For an analysis of shared and unique taxa between different samples, jvenn2 was used [36]. Differential abundance was plotted using GraphPad Prism 8 software to determine statistical significance using Mann–Whitney and *p* value with Bonferroni correction as the number of samples was uneven. Microbial genera with *p* value < 0.005 were considered significant. For the comparative phylogenetic analysis, the reads of all 8 samples analyzed in this study along with six samples of Chaudhari et al. (CH1–CH6) [18], one sample each of Bisht et al. (BI) [19], and Rathour et al. (RA) [16] were classified through Kraken2 using RDP (RDP database version 2020). The obtained classified reads output was concatenated using Pavian [33] and used as an input in MEGAN (MEGAN CE V6.17.0) to visualize the phylogenetic tree along with the quantitative data of each taxa.

## 3. Results and Discussion

### 3.1. Enumeration of Bacteria on Microbiological Media

Table 1 highlights CFU obtained on different microbiological media. Significant variations in the CFU were observed among the different media, pH, and sample types. The CFU/mL varied from 3 × 10^3^–8.6 × 10^4^ in the water sample to 2.3 × 10^4^–3.1 × 10^5^ CFU/g in the sediment sample. At pH 8.0, R2A agar supported the highest number of colonies for both water (8.6 × 10^4^ CFU/mL) and sediment (3.1 × 10^5^ CFU/g) samples, whereas at pH 9.0, R2A100 times diluted medium recovered the maximum number of colonies in the water (6.8 × 10^4^ CFU/mL) and sediment (7.8 × 10^4^ CFU/g) sample (Table 1).

### 3.2. Physico-Chemical Analysis

Pangong lake is a brackish water, high-altitude lake, which remains frozen during winter for three months (December, January, and February, Bhat et al. [19]). The lake ecosystem faces extreme environmental conditions such as high salinity, alkalinity, electrical conductivity, and temperature in which microbes are able to survive with adaptive strategies [37]. At 0.12%, a negligible amount of TOC was detected, thus reflecting the oligotrophic conditions of the ecosystem. Both sediment and water samples were alkaline with respective pH values measured as 8.32 (1.565 mΩ/cm electrical conductivity) and pH 8.0–9.0 (Appendix A), confirming the earlier findings of Bhat et al. [19], Rathour et al. [16], and Chaudhari et al. [18]. Metals such as calcium, magnesium, zinc, manganese, iron, boron, copper, cobalt, molybdenum, cadmium, chromium, aluminum, barium, nickel, lead, mercury, arsenic, chloride, lithium, vanadium, selenium, silicon, phosphorus, sulphate, ammoniacal nitrogen, nitrate nitrogen, and nitrite in the sediment sample were within permissible limits (Appendix A) as per the WHO guidelines (https://www.omicsonline.org/articles-images/2161-0525-5-334-t011.html, accessed on 3 May 2023) and Alloway, B. J. [38].

### 3.3. Analysis of Bacterial Diversity by 16S rRNA Gene-Based Tag Sequencing

In traditional culturable techniques, only replicating cells that can form visually well-separated colonies can be sub-cultured, whereas abiotic parameters, i.e., temperature, pH, UV, nutrients, and absence of cell-to-cell communication may affect bacterial growth by forcing cells into dormancy, called the viable but non-culturable (VBNC) state [39,40]. These microbes are metabolically active but cannot form colonies on conventional media and grow only under favorable environmental conditions [39,41]. Further, it is a well-established fact that no bacteria types from an environmental sample can be cultured and identified [25,39,42], and conventional culturing processes are labor-intensive as well as logistically and economically intensive; therefore, in this study, the direct sequencing of colonies/cells through long read technology grown on media samples (i.e., total CFU on R2A, R2A100, and TSBA500) was compared with direct samples (PW, PSD), wherein total DNA was extracted from water and sediment samples directly to ascertain the effect of media formulations on the culturability pattern and to compare their effectiveness in retrieving the in situ diversity (Figure 1). Moreover, since the majority of the reads analyzed in all the samples were nearly full length (Appendix A), the identification of classified reads at the genus level could be achieved with more confidence.

A total of 230,129 reads were obtained in the range of 11,795 to 45,633 (Appendix A). Among the direct samples, i.e., PSD and PW, 100% of reads were classified, whereas for the media samples, these were in the range of 100% to 69.1% (Appendix A). The classified reads belonging to the phyla Pseudomonadota, Bacteroidota, Cyanobacteria, Bacillota, Verrucomicrobiota, and Actinomycetota were shared by both PW and PSD samples, while the phyla Planctomycetota, Deinococcus-Thermus, Acidobacteriota, Gemmatimonadota, Hydrogenedentes, and Nitrospirota were only detected in the PSD sample, indicating that the sediment was most diverse (Figure 2; Appendix A).

It has been generally observed that sediments host a diverse bacterial composition due to the complex and heterogeneous gradients of substrate, pH, and redox potential, forming several microhabitats consisting of a vast matrix of organic and inorganic solid surfaces for bacterial growth [43,44,45]. The phyla Bacteroidota, Actinomycetota, and Cyanobacteria were relatively more abundant in PW and PSD compared to culture media samples (Figure 2; Appendix A). Cyanobacteria were detected in direct samples (PSD, PW) and recovered only in diluted media (PW_R2A100, PW_TSBA500), indicating that low nutrient conditions were mandatory for their isolation. They are involved in key roles as autotrophic primary producers in carbon and nitrogen cycles through photosynthesis and nitrogen fixation, forming the base of the aquatic food chain and maintaining the ecology of the lake ecosystem [46,47]. Interestingly, for both the water and sediment samples, diluted media seemed to capture a higher diversity at the phyla level, which is evident by the detection of Actinomycetota in the sediment samples and Cyanobacteria in the water samples (Appendix A). Some of the bacterial groups, i.e., Planctomycetota, Verrucomicrobiota, Acidobacteriota, and Gemmatimonadota, were recovered only in the direct samples (PSD and PW) and not in any of the culture media, with the latter three phyla detected only in the sediment sample (PSD; Appendix A). It is a well-known fact that some of these taxa form part of the PVC superphylum and are extremely slow growing, very difficult to culture (require special media compositions), and several representatives of these lineages have only been recovered as metagenome assembled genomes (MAGs) [48]. It is also pertinent to note that Bacillota and Bacteriodetes were over- and under-represented, respectively, in the majority of the cultured media, thus undermining the biasness of the axenic media towards certain groups (Appendix A). The Deinococcus-Thermus group detected only in the PSD sample withstands high temperatures and extreme UV radiation [49] as the Pangong Tso lake undergoes frequent temperature shifts and high UV radiation [7] due to its location at an extremely high altitude. Similar patterns of diversity with Pseudomonadota as the dominant phylum followed by Bacteroidota have been reported from sediment samples of the lake, although there is substantial variation in terms of the relative abundance of these taxa, which might be because of different sequencing and sampling methods [16,18].

At the class level, Gammaproteobacteria was most predominant in sample PW, whereas both Gamma- and Betaproteobacteria dominated the PSD sample, with the relative abundance of Alphaproteobacteria being much higher in PW compared to PSD, while Deltaproteobacteria was unique to the PSD sample (Appendix A). It was interesting to observe this distribution pattern as previous reports of the lake have indicated a much higher abundance of either Gammaproteobacteria [16] or Gammaproteobacteria and Alphaproteobacteria [18], with Betaproteobacteria represented as a minor population. Abundant genera (≥1%) in PSD sample were *Hydrogenophaga* (~15.7%), *Acinetobacter* (4.6%), *Pseudomonas* (4.5%), *Arenimonas* (2.1%) followed by *Comamonas*, *Stenotrophomonas*, *Rheinheimera*, *Methylotenera, Opitutus, Truepera*, *Rhodoferax*, *Variovorax*, *Burkholderia*, *Delftia*, *Acidovorax*, *Thiobacillus*, and *Lysobacter*, while the top five genera in the PW sample were *Rheinheimera* (20.7%), *Pseudomonas* (15.3%), *Loktanella* (7.5%), *Marinomonas* (5.4%), and *Flavobacterium* (4.1%) with a moderate abundance of *Escherichia/Shigella, Belliella*, *Providencia*, *Alishewanella*, *Sulfitobacter*, *Paracoccus*, *Serratia, Planktomarina, Enterobacter*, *Roseovarius*, *Owenweeksia*, and *Klebsiella* (Figure 3 and Appendix A; Appendix A).

The study of shared and unique taxa showed that PSD and PW samples contained 42 core genera, i.e., present in both water and sediment. The top five core genera were *Rheinheimera*, *Pseudomonas*, *Hydrogenophaga*, *Loktanella*, and *Acinetobacter* (Figure 4A). The analysis suggested that sediment samples contained a higher number of unique taxa (82), almost 50% more than the PW (40) sample (Figure 4B, Appendix A).

The *Hydrogenophaga* spp. are facultative chemolithoautotrophs and majorly grow by the oxidation of organic compounds, but few species can oxidize H_2_/CO-as an energy source and utilize CO_2_ as a C source growing mixotrophically, and some have the capability to fix N_2_ and reduce NO_3_ under anaerobic conditions [50]. None of the previous reports, however, have identified this taxon from the sediment/water samples of the lake as an abundant taxa [14,15,16,17,18,19]; however, its versatile metabolic capabilities make it an interesting candidate for further investigation (Appendix A) [51,52]. It is important to note that in culture-based studies, several strains of the genus have been isolated in our lab and few seem to be novel species after preliminary characterization (unpublished data). The genus *Loktanella* of the *Roseobacter* group within the family *Rhodobacteraceae* in Alphaproteobacteria is a group of chemoorganoheterotrophic bacteria mostly isolated from marine environments with few species harboring bacteriochlorophyll a pigment and pufLM genes [53]. However, molecular studies targeting photosynthetic reaction centers (pufLM) and amplicon sequencing have revealed an abundance of *Loktanella*-like sequences in the saline lakes of the Tibetan plateau [54] and within the bacterial community of glacio-marine systems in the Arctic region [55]. Interestingly, Chaudhari et al. [18] have reported nearly half (49%) of the total bacterial community in the Pangong water sample as belonging to *Loktanella* (Appendix A). Although the authors described these sequences as belonging to aerobic anoxygenic phototrophic (AAP) bacteria, one needs to be cautious as none of the validly described species have an established role in photosynthesis and phylogenetic identity and the function of *Loktanella*-like sequences detected by Jiang et al. [54] have to be validated through more robust methods like whole-genome, metagenome, and culture-based approaches. The genera *Pseudomonas*, abundant in both lake sediment and water samples (~4.5% and 15% in PSD and PW, respectively), are strictly aerobic chemo-organotrophs that are ubiquitous in their distribution and have the ability to oxidatively degrade a variety of substrates including alkanes and aromatic compounds [56]. Many species are psychrotrophic, denitrifiers, and few have the ability to grow in low nutrient conditions [56,57]. Previous reports [16,18] have identified the group as an abundant part of the microbial community only in the sediment samples (Appendix A). *Rheinheimera* spp. highly abundant in the PW sample (~20%) and moderate in PSD (~2%) are chemoheterotrophic Gammaproteobacteria, growing optimally at salinity levels of 3% (*w*/*v*), rapidly degrade organic matter, and play a critical role in the biogeochemical cycling of carbon [58,59,60,61]. Importantly, previous culture-based [61,62,63] and functional metagenomic surveys [16] have identified this group as an important part of the microbial community of haloalkaline lakes (including Pangong) with the identification of genes related to ammonia, inorganic sulfur assimilation, and antimicrobial production [62]. The other two abundant taxa in the water sample, i.e., *Marinomonas* and *Alishewanella* within Gammaproteobacteria, belong to the families *Oceanospirillaceae* and *Alteromonadaceae* (recently transferred to the family *Chromatiaceae* [64]) that consist mainly of aerobic/facultatively anaerobic heterotrophs of mainly marine origin that require Na+ for growth with potential applications in the production of antimicrobials, melanin, and dye degradation, respectively [65,66]. Some of the *Alishewanella* spp., *A*. *longhuensis* [60], and *A. alkalitolerans* [64] have, in fact, been isolated from similar haloalkaline lakes, and it seems that soda lakes share similar compositions of the bacterial community with marine habitats in terms of the identity of few *Pseudomonadota* groups. Therefore, it may be prudent to use growth media like marine agar, etc., in future to target the isolation of native bacteria from such lake ecosystems since most of these microbes have an optimum Na^+^ requirement in the range of 3% (*w*/*v*) to maintain high osmotic pressure due to the high Cl^−^ anion in the lake sediment samples (Appendix A; [16,18]). This is a major contributor determining the microbial community composition [4,67,68]. Within the phylum Bacteroidota, *Flavobacterium* spp. were the most abundant (4%) in the water sample (PW, Appendix A). Members are mostly psychrotolerant with optimum growth at 20–30 °C, abundant in cold aquatic ecosystems such as lakes, rivers, streams, etc. [69], and they degrade a variety of proteins and polysaccharides of algae, plants, fungi, insects, etc. [70,71,72,73,74,75]. Rathour et al. [16] and Chaudhari et al. [18], however, identified this taxon only in the sediment samples. The next abundant genus within the phylum was *Belliella* (Appendix A), which are chemoheterotrophs isolated from alkaline, saline, and thermal habitats. In fact, three species, i.e., *Belliella kenyensis* [76], *B. aquatica* [77], and *B. buryatensis* [78] have been isolated from haloalkaline lakes located in Kenya, China, and the Republic of Buryatia (Russia), respectively. All the species described to date have optimum pH and salinity in the alkaline range and around 3% NaCl, respectively [78].

Among the moderately abundant taxa (~≤2%) in the sediment samples, *Acidovorax, Burkholderia*, *Comamonas*, *Delftia*, *Rhodoferax*, and *Variovorax* spp. of the order *Burkholderiales* within *Betaproteobacteria* are widely distributed in natural and man-made environments and are diverse in their metabolism. *Rhodoferax* spp. uses acetate, pyruvate, lactate, and succinate as the sole carbon sources [79]. The genera *Variovorax*, *Delftia*, and *Comamonas* spp. are capable of accumulating polyhydroxyalkanoates [80] and have a distinctive catabolic pattern involving protocatechuate meta- and ortho-cleavage pathways [81]. Some *Variovorax* spp. are chemoorganotrophs, capable of using a wide range of organic compounds for growth. *Variovorax* and *Comamonas* spp. were reported as minor populations in the Pangong sediment sample [18]. *Burkholderia* spp. display associative nitrogen fixation (nodule formation) in soil or sediment [79]. A Betaproteobacterium member, *Methylotenera* belonging to family *Methylophilaceae*, was observed only in the PSD sample (Appendix A). *Methylotenera* spp. are methylotrophs and use one carbon compound, methane or methanol, for their growth as a carbon source. Methylotrophs are mainly distributed in marine or brackish habitats [17], and previous studies of this ecosystem have reported the detection of *Methylotenera* in soil water mix collected from Pangong lake. Furthermore, *Methylococcaceae, Methylothermaceae*, *Methylophilaceae*, *Methylobacteriaceae*, and *Methylocystaceae* in a Pangong lake sediment sample were reported by Rathour et al. [16], wherein they identified *Methylophaga* as a major population (>10%, Gammaproteobacteria). Collectively, the facts suggest that methylotrophs are abundant in Pangong lake and play an important role in the methane cycle. Future functional metagenomic studies targeting methane metabolism can shed useful insights into the roles and mechanisms of methylotrophs in relation to C cycling. The detection of *Escherichia/Shigella*, *Klebsilla*, *Serratia*, *Enterobacter*, and *Providencia* spp. of the family *Enterobacteriaceae* in the water samples indicates the contamination of the lake by anthropogenic activities. Pangong lake is one of the tourist hot spots in Ladakh, India; thus human interference might affect its water quality.

With regard to our second objective to understand the effect of media formulation in terms of supporting the growth of a wide variety of species, analyses of diversity indices indicated that both R2A- and R2A100-times diluted media supported better species richness for the sediment sample with almost comparable values for Shannon, Simpson, chao-1, and total number of taxa, whereas for the water sample, the corresponding values of 2.306, 0.7624, 314.6, and 233 were highest for the R2A media (Table 2).

Although R2A100 showed a slightly higher chao-1 index, one must take into account the fact that it is a non-parametric-based index and takes into consideration rare taxa, whereas the evenness-based Simpson index for the PW_R2A sample was much higher compared to PW_R2A100 (0.7624 vs. 0.2601, Table 2). The rarefaction curves revealed the requirement of more sequencing depth to cover the scale of diversity for PSD and PW samples, whereas all the media sample curves showed saturation, indicating good coverage (Appendix A). The phylum Pseudomonadota was the most abundant in direct and media samples. This ubiquitous predominance has been observed in different freshwater lakes of the Himalayan region as well as globally, especially in molecular surveys [82]. Bacteroidota was the second most abundant phylum in PSD, PW, PSD_R2A, and PSD_R2A100 samples, while Bacillota took this spot in media samples PSD_TSBA500, PW_TSBA500, PW_R2A, and PW_R2A100, highlighting the selective biasness of commonly used growth media in the over-representation of a few taxa in culturable diversity surveys. In fact, previous reports by Sahay et al. [15] and Yadav et al. [14,83] recovered bacterial groups belonging to Bacillota as the dominant population. However, none of the media could retrieve Planctomycetota, Verrucomicrobiota, Deinococcus-Thermus, Acidobacteriota, and Gemmatimonadota, indicating that the incubation time/culture media conditions were insufficient/inappropriate, respectively, for cultivation. A comparison of shared and unique taxa between total DNA samples (PSD and PW) and media samples was made in order to understand the number of taxa that can be retrieved in media. In the sediment sample, a total of 34 genera were detected in both PSD and PSD media samples, wherein *Pseudomonas* and *Hydrogenophaga* were the top two genera (Figure 4C). However, 90 genera were exclusively detected in the PSD sample, which could not be retrieved on any media samples, and only 10 genera were unique in media samples (Figure 4D, Appendix A). In the case of water samples, 24 genera were core taxa in PW and PW media samples, whereas the top 2 genera were *Pseudomonas* and *Rheinheimera* (Figure 4E). A total of 58 genera were exclusively detected in PW samples that could not be retrieved on any media samples, whereas 6 genera were uniquely detected in PW media samples (Figure 4F, Appendix A). The differential abundance analysis between sediment and water, considering all the samples, i.e., total DNA and media samples, revealed that the genera *Hydrogenophaga* and *Pseudomonas* were significantly associated with sediment and water samples, respectively. Although statistically non-significant, the genera *Acinetobacter, Arenimonas*, *Rheinheimera, Escherichia, Loktanella, Marinomonas,* and *Flavobacterium* were associated with sediment and water samples, respectively (Appendix A). Among Gammaproteobacteria, the genera *Pseudomonas* was observed in all the samples, while *Rheinheimera* and *Flavobacterium* were detected in all the samples except PW_TSBA500 and PSD_R2A, respectively. Some genera such as *Methylotenera*, *Thiobacillus*, *Lysobacter*, *Opitutus*, *Truepera*, *Gemmatimonas*, *Bradymonas*, *Ilumatobacter*, *Marinobacter, Seohaeicola*, and *Reinekea* were detected only in direct samples (PW and PSD; Figure 3, Appendix A), whereas *Roseovarius, Sulfitobacter, Alishewanella*, *Loktanella*, *Planktomarina*, *Ruegeria*, *Porphyrobacter*, *Novosphingobium*, *Erythrobacter*, *Staphylococcus,* and *Sphingobium* were observed only in diluted media samples and direct samples (Figure 3, Appendix A). The PCA plot represented a correlation of the genera *Rheinheimera*, *Pseudomonas*, *Escherichia/Shigella*, *Providencia*, etc., with water sample (PC3), whereas *Hydrogenophaga*, *Bacillus*, *Thiobacillus, Algoriphagus*, *Methylotenera*, etc., grouped with sediment samples (PC1) with cosmopolitan bacterial genera, i.e., *Pseudomonas*, and *Bacillus*, were over-represented especially in diluted media, indicating selection biasness in growth media and culture conditions (i.e., incubation times) (Appendix A). Interestingly, some of the abundant taxa, such as *Hydrogenophaga*, *Pseudomonas,* and *Rheinheimera* spp., were also recovered in our culture-dependent analysis, representing 3.5%, 25%, and 1.4% of the total isolates identified, respectively (Appendix A). In contrast, moderately abundant OTUs belonging to *Acidovorax*, *Burkholderia*, *Delftia*, *Limnohabitans*, and *Rhodoferax* were better represented in media samples, although none could be recovered in culture indicating the inherent procedural flaws in our conventional approach (Appendix A). From the culturable work, we were able to isolate and identify a few facultative methylotrophs, i.e., *Methylobacterium* spp. (classified reads retrieved in PSD sample) and *Methylorubrum* spp., which are known to use C1 compounds like methanol and methylamines. This is not surprising given that the lake sediment contains a variety of trace elements, including Fe, Co, Mo, Mg, Mn, and Zn, as well as nitrogen, which is available as nitrate, nitrite, and ammonia, promoting the growth of such bacterial groups (Appendix A). Interestingly, classified reads affiliated to genera *Enterococcus, Jonesia*, *Streptophyta*, *Reyranella*, *Arthrobacter*, *Blastomonas*, *Gemmobacter*, *Mesorhizobium*, *Pseudorhodobacter*, *Ensifer*, *Haematobacter*, *Roseinatronobacter*, *Tabrizicola*, *Phyllobacterium*, *Aureimonas*, *Methylarculla*, *Ochrobactrum*, *Sphingorhabdus*, and *Nitrosospira* were detected only in media samples, suggesting that these taxa might be less abundant in situ and may become enriched in growth media, as evident by the isolation of several strains of *Tabrizicola* spp., *Blastomonas* spp., and *Jonesia* spp. in our lab (Appendix A, Figure 3). Previous reports on culturable diversity have identified *Arthrobacter*, *Pseudomonas*, *Bacillus*, *Planococcus*, *Exiguobacterium*, *Paenibacillus*, *and Staphylococcus* spp. as dominant groups involved in the production of several polymer degrading-enzymes [14,15,81]. It is pertinent here to mention that *Planococcus antartcticus* and *Paenibacillus xylanexedens*, reported previously, were successfully re-isolated and their functional activity was confirmed in our analyses, indicating niche specificity.

## 4. Conclusions

The present study attempted a comparison of the total and culturable bacterial diversity from high-altitude Pangong lake using long read sequencing technology, i.e., Oxford Nanopore, to understand the microbial composition within the saline high-altitude lake water and sediment samples. Our study indicated that the sediment hosted the most diverse bacterial community. Phyla Pseudomonadota, Bacteroidota, Cyanobacteria, Bacillota, Verrucomicrobiota, and Actinomycetota were observed both in sediment and water samples, while Planctomycetota, Deinococcus-Thermus, Acidobacteriota, Gemmatimonadota, Hydrogenedentes, and Nitrospirota were exclusive in the sediment. Many ecologically important genera like *Hydrogenophaga*, *Loktanella*, *Pseudomonas*, *Rheinhemera*, *Flavobacterium, Marinomonas*, *Alishewanella*, *Acinetobacter*, *Arenimonas*, and *Comamonas* were abundantly observed in both sediment and water samples of the ecosystem. The genera *Hydrogenophaga*, *Rheinheimera*, and *Loktanella* seemed to be unique and abundant taxa for the sediment and water samples, respectively. *Loktanella* appears to be an endemic microbial population as established in previous reports [18]. Some of the functionally important groups, i.e., *Methylotenera*, that play an important role in the C cycle through the utilization of C1 compounds were detected exclusively in the sediment. The presence of such phenotypic groups has been confirmed in previous metagenomic surveys of the habitat. The detection of Enterobacteriaceae in the water samples indicates contamination through anthropogenic activities.

In terms of culturability, R2A diluted 100 times and undiluted R2A seemed best suited for recovering bacterial diversity from sediment and water samples, respectively. Bacteria belonging to Planctomycetota, Verrucomicrobiota, Deinococcus-Thermus, Acidobacteriota, Gemmatimonadota, Hydrogenedentes, and Nitrospirota could not be detected in any of the culture media. One of the reasons for this was that culture conditions/timelines employed for isolation do not correspond with the above groups. Further, it might also be possible that the DNA extraction method could not lyse the cell walls of some groups (i.e., Deinococcus-Thermus) efficiently. It is pertinent to mention that future large scale culturing efforts could use marine agar as the majority of the abundant taxa detected have 3% salinity requirement for optimal growth. In recent years, through the advent of genome sequencing technologies, Actinomycetes and Cyanobacteria have been shown to encode the largest biosynthetic diversity [84], and diluted/low-strength media could be utilized for targeting the cultivation of novel lineages within these taxa.

## Figures and Tables

**Figure 1 microorganisms-12-00397-f001:**
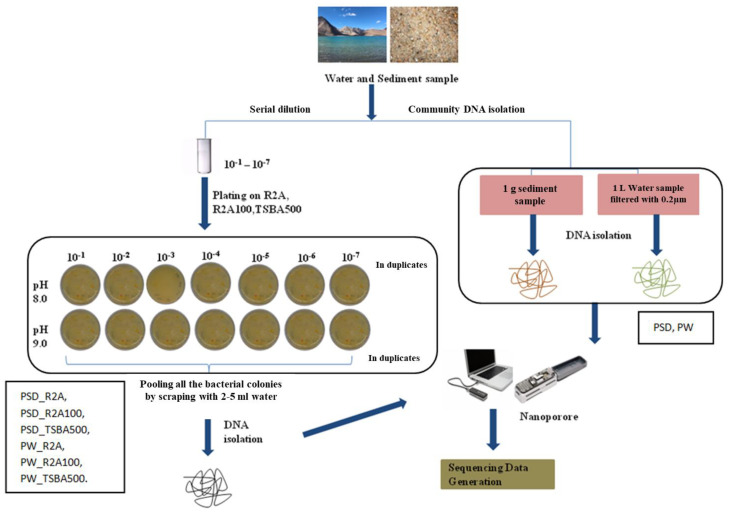
Flowchart representing strategy of microbial diversity analysis of Pangong lake water and sediment samples.

**Figure 2 microorganisms-12-00397-f002:**
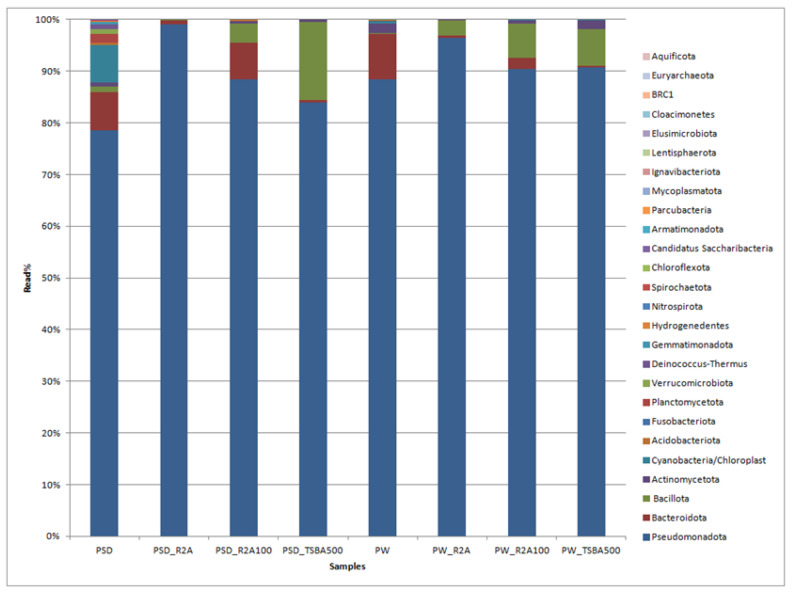
Bar graph displaying the taxonomic distribution of bacterial diversity at the phyla level in direct and media samples.

**Figure 3 microorganisms-12-00397-f003:**
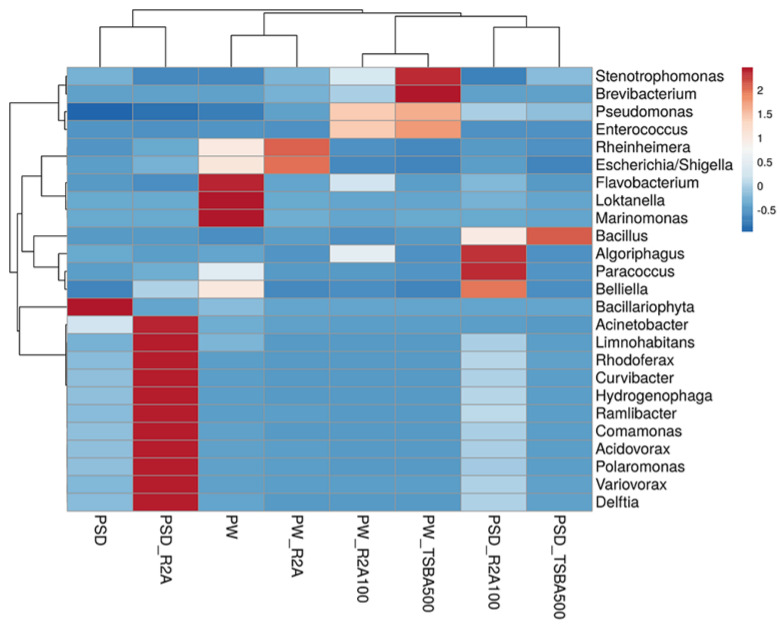
Heatmap showing distribution of genera-level diversity in both direct samples and media samples.

**Figure 4 microorganisms-12-00397-f004:**
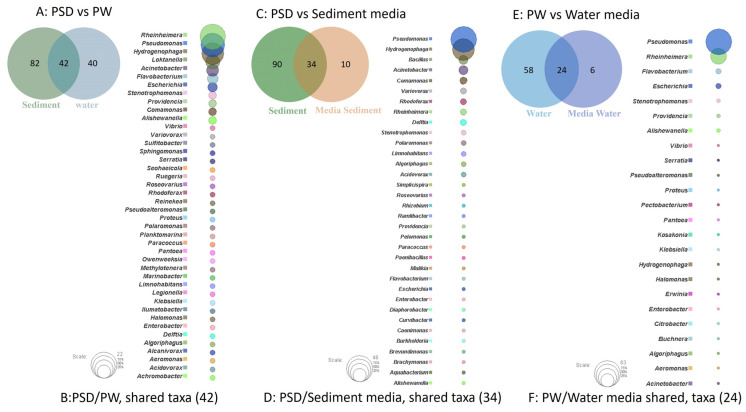
Venn diagram representing details of total number of shared and unique taxa (at genus level) among sediment (PSD) and water (PW) samples (**A**); PSD and PSD media samples (**C**); and PW and PW media sample (**E**). Bubble plot depicts the identity of shared taxa among sediment (PSD) and water (PW) samples (**B**); PSD and PSD media samples (**D**); and PW and PW media sample (**F**).

**Table 1 microorganisms-12-00397-t001:** Average CFU of water and sediment samples on different media at pH 8.0 and pH 9.0.

**Media Dilutions**	**Pangong Water (PW)** **pH 8.0 (CFU/mL)**	**Pangong Water (PW)** **pH 9.0 (CFU/mL)**
**R_2_A**	8.6 × 10^4^	4.5 × 10^4^
**R_2_A_100_**	3.0 × 10^3^	6.8 × 10^4^
**TSBA_500_**	3.0 × 10^4^	2.6 × 10^4^
**Media Dilutions**	**Pangong Sediment (PSD)** **pH 8.0 (CFU/g)**	**Pangong Sediment (PSD)** **pH 9.0 (CFU/g)**
**R_2_A**	3.1 × 10^5^	2.6 × 10^4^
**R_2_A_100_**	1.0 × 10^5^	7.8 × 10^4^
**TSBA_500_**	1.5 × 10^5^	2.3 × 10^4^

**Table 2 microorganisms-12-00397-t002:** Diversity indices of bacterial diversity of direct and media samples.

	PSD	PSD_R2A	PSD_R2A100	PSD_TSBA500	PW	PW_R2A	PW_R2A100	PW_TSBA500
**Taxa_S**	532	338	371	155	453	233	202	196
**Individuals**	8994	29,701	19,785	10,641	15,371	15,076	25,651	28,611
**Dominance_D**	0.04969	0.2004	0.2305	0.5653	0.08166	0.2376	0.7399	0.7083
**Simpson_1-D**	0.9503	0.7996	0.7695	0.4347	0.9183	0.7624	0.2601	0.2917
**Shannon_H**	4.286	2.63	2.646	1.181	3.613	2.306	0.746	0.7683
**Evenness_e^H/S**	0.1366	0.04106	0.03799	0.02101	0.08184	0.04307	0.01044	0.011
**Chao-1**	727.5	534.9	537.4	330.6	699.1	314.6	416	610.4

## Data Availability

All data were submitted to the NCBI Sequence Read Archive (SRA) under the following accession numbers: SRR11354751 (PW)-SRR11354747 (PSD); SRR11354746 (PSD_R2A); SRR11354745 (PSD_R2A100); SRR11354744 (PSD_TSBA500); SRR11354750 (PW_R2A) SRR11354749 (PW_R2A100); and SRR11354748 (PW_TSBA500).

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
