# Peer review of "Analysis of Culturable Bacterial Diversity of Pangong Tso Lake via a 16S rRNA Tag Sequencing Approach"

_microorganisms, 2024, doi:10.3390/microorganisms12020397_

Round 1

Reviewer 1 Report

Comments and Suggestions for Authors

The manuscript entitled ‘Analysis of culturable bacterial diversity of Pangong Tso lake by metagenomics approach’ analyzed the diversity of the culturable bacteria by applying a high throughput amplicon sequencing approach based on long read technology to determine the spectrum of bacterial diversity supported by axenic media based samples collected from Pangong Tso lake. It is good, however, I don’t think it is suitable for publication in Microorganisms. The biggest flaw is the lack of novelty. As the author describes in the introduction, some similar works have been carried out. This work did not provide a significant breakthrough. It should highlight the importance of the work by comparing the previous results. The second flaw was the number of sample sites which was not enough, as the author displayed, ‘Samples (both sediment and water) 65 were collected in the month of September 2017 in 50 ml sterile falcon tubes (Tarsons, Fig-66 ure S1) and whirl pack bags (1L, Hi-Media).’. I don’t think the samples collected were representative. I suggested the author expand the sample collection region and compare the characteristics of the microbial community. And there are some mistakes which should be improved.

1.Some genera should be italicised. For example, in line 283, the Flavobacterium should be ilatic. Similar to this problem, the author should correct the associated problems.

2.Some spelling mistakes, for example, in line 360, I don’t understand the meaning of OUT. (line 360, ‘In 359 contrast, moderately abundant OUT’s belonging to Acidovorax, Burkholderia, Delftia, Lim-‘).

Comments on the Quality of English Language

no

Author Response

Comments and Suggestions for Authors

  • The manuscript entitled ‘Analysis of culturable bacterial diversity of Pangong Tso lake by metagenomics approach’ analyzed the diversity of the culturable bacteria by applying a high throughput amplicon sequencing approach based on long read technology to determine the spectrum of bacterial diversity supported by axenic media based samples collected from Pangong Tso lake. It is good, however, I don’t think it is suitable for publication in Microorganisms. The biggest flaw is the lack of novelty. As the author describes in the introduction, some similar works have been carried out. This work did not provide a significant breakthrough. It should highlight the importance of the work by comparing the previous results.

Ans:  Thank you for your query. Yes the analysis of bacterial diversity from Pangong lake has been conducted before by Bisht et al., 2018, Chaudhari et al., 2020, and Rathour et al., 2020 by culture independent approach and Yadav et al., 2015 by culture dependent approach. Here in the present study first time, we did comparative bacterial diversity analysis to ascertain the differences in composition of bacterial communities determined from direct samples (PSD & PW) and media samples (PW_R2A, PW_R2A100, PW_TSBA500, and PSD_R2A, PSD_R2A100 and PSD_TSBA500). This type of approach has been used previously only once in a report by Dyda et al., 2017 to choose microbiological mediums that facilitate the growth of bacteria in a manner that closely resembles the proportions and diversity found similar to the in-situ microbiomes of historical stones. We have addressed the advantages of this approach in our detailed response to query from reviewer 2. Pl also see our reply to query no 11 of reviewer 3 and the relevant manuscript text in results and discussion section (lines 254-262) pertaining to identification of the taxa Hydrogenophaga as an abundant taxa for the first time from this lake ecosystem although none of the previous culture-dependent and metagenomics (both whole genome and 16s tag sequencing) could detect this group. This fact was also established through isolation of several novel strains of Hydrogenophaga in our culturing approach (see our response to query 8 of reviewer 3). Thus we respectfully disagree with the reviewer in concluding that the manuscript lacks novelty.

  • The second flaw was the number of sample sites which was not enough, as the author displayed, ‘Samples (both sediment and water) 65 were collected in the month of September 2017 in 50 ml sterile falcon tubes (Tarsons, Fig-66 ure S1) and whirl pack bags (1L, Hi-Media).’. I don’t think the samples collected were representative. I suggested the author expand the sample collection region and compare the characteristics of the microbial community. And there are some mistakes which should be improved.

Ans: The lake covers a large geographic area and during sampling there were restrictions imposed by the military in terms of collecting samples from different locations as the site is quite close to the India-China border. Both water and sediment samples were collected from nearby different portions and pooled for a good representation. Indeed, it would have been better to analyze more number of samples, but as we were also interested to conduct a culture based study with different media formulations we restricted our sample size due to economics of the whole process. Further also note our response to query 6 of reviewer 3 in terms of isolating a representative community DNA and query 8 wherein we have included a comparative analyses between the sequencing and culturing approach in the revised version. We would like to mention, however, that we did collect samples from nearby ecosystems of the lake habitat (upstream river, shyok river and rhizosphere sample of seabuckthorn plant) whose results are not part of this manuscript as only the lake ecosystem community composition has been depicted in this study.

Keeping in perspective the difficult terrain of sampling it would not be possible to collect additional samples for this study.

  • Some genera should be italicised. For example, in line 283, the Flavobacterium should be ilatic. Similar to this problem, the author should correct the associated problems.

Ans: Thank you for your corrections, we have corrected as per your suggestion in line no 14, 16, 17, 197, 204, 223, 226, 231, 233, 279, 306, 320, 341, 383, 384, 396, 441, and 453.

  • Some spelling mistakes, for example, in line 360, I don’t understand the meaning of OUT. (line 360, ‘In 359 contrast, moderately abundant OUT’s belonging to Acidovorax, Burkholderia, Delftia, Lim-‘).

Ans: We are sorry for the typing error; this has been corrected to OTU’s instead of OUT’s in line no 401.

Reviewer 2 Report

Comments and Suggestions for Authors

The article aims to develop a new approach to the study of structural and functional microbial diversity, based on combining high-throughput next-generation sequencing with traditional cultivation methods. This topic is quite relevant for the search for new strains suitable for use in biotechnology. The article is well prepared and structured. However, despite the relevance and great work done by the authors, I had a comment, and therefore I recommend a minor revision.

In my opinion, the authors did not reveal the advantage of their approach compared to traditional ones: what new did this approach provide, except for the assumption of using marine agar for subsequent cultivations?

Author Response

Comments and Suggestions for Authors

The article aims to develop a new approach to the study of structural and functional microbial diversity, based on combining high-throughput next-generation sequencing with traditional cultivation methods. This topic is quite relevant for the search for new strains suitable for use in biotechnology. The article is well prepared and structured. However, despite the relevance and great work done by the authors, I had a comment, and therefore I recommend a minor revision.

In my opinion, the authors did not reveal the advantage of their approach compared to traditional ones: what new did this approach provide, except for the assumption of using marine agar for subsequent cultivations?

Ans: Thank you for your query. We believe that some of the biggest challenges in traditional cultivation approach is inability to cultivate majority of the bacteria especially novel strains and the logistics of cost and labor considering high-throughput sample processing.

The approach described in the manuscript offers a practical approach for deciphering optimized media formulations in terms of retrieving the best possible culturable taxa compared to in-situ samples targeting a specific bacterial group, in this case heterotrophic bacteria. The approach allows faster identification of taxa (classified reads) from samples by plating on different types of media through long read technology that retrieves and accurately identifies the OTUs correctly closest to cultured strains because of full length 16S sequencing. This point has been well highlighted in the first para of section 3.3 of  revised manuscript (lines no 167-182). Further interestingly in our analysis we detected few taxa i.e. Hydrogenophaga, Burkholderia, Delftia, Limnohabitans, and Rhodoferax abundantly which were earlier not reported either in any culture based (Yadav et al., 2015; Sahay et al., 2012) or metagenomics (both whole genome metagenome by Rathour et al., 2017, 2020 and 16S tag sequencing approach by Bisht et al., 2018; Chaudhari et al., 2020) of the same habitat which supports the robustness of the approach. Therefore data obtained from such analyses can be utilized in selecting/designing specific media for obtaining in culture these novel lineages/groups that till date have not been detected in this lake ecosystem. We would also like to add that this study was limited to only to heterotrophic bacteria applying few media formulations. Several important physiological groups i.e Cyanobacteria, phototrophic bacteria and Diazotrophs etc were not targeted. This could be the next objective and we may in the process identify as yet not reported novel lineages from the same ecosystem. Thus in our opinion the approach has far reaching implications and potential to decide the best media formulation for an ecological niche targeting a specific population.

Reviewer 3 Report

Comments and Suggestions for Authors

The manuscript “Analysis of culturable bacterial diversity of Pangong Tso lake by metagenomics approach” by Yadav et al. presents the analysis of the diversity of the culturable bacteria by applying a high throughput amplicon sequencing approach based on long read technology to determine the spectrum of bacterial diversity supported by axenic media.

The title is a bit misleading: metagenomics approach is not what the authors results provided, which was basically amplicon sequencing data based on the full length 16S rRNA gene (1.5Kbp covering regions V1-V9). Please correct accordingly!

Line 4: The name of the corresponding author is missing; or at least “and” is misleading…

Materials and Methods:

2.1. Sample collection: lines 67-68: The authors mentioned that the samples were transported to the laboratory within 48h after collection. This is quite a long time.. How were the samples stored in-between? There is a lot of research around confirming that even hours (of “unproper” storage) are enough to change the community composition, but definitely to affect the amount of detectable nutrients or carbon availability. It is highly crucial to keep the samples in the dark, and cold; in a “perfect world” directly frozen..., otherwise the community or nutrient composition might have changed...

Another issue would be the composition of the used media for cultivation (please provide a table for the media components) and were the plates cultivated in the dark or light; since many bacteria are capable of anoxygenic photosynthesis?

Was the analysis of the subsequent samples (especially PSD and PW) performed in replicates (preferably three independent DNA isolations)?

In general, the author should provide a separated view on the communities which were originally present in the sediment and the surrounding water (and present them in a Figure (or Table), maybe containing more information than just the phyla level) and then compare to the obtained communities via cultivation different media. Provide the data analysis in the main manuscript and not in the supplemental table. How many species could actually be retrieved by the culturing attempts? It could be a valuable information to get deeper insight into the bacterial diversity of Pangong Tso not just on phyla level and only for bacteria, but also cyanobacteria on class, family or genus levels. And maybe compare the most abundant taxa, especially with previous? Would there be (m)any correlations?

There is no need to cite permanently the literature twice, since it interrupts the reading flow, e.g.  Line 151-152, but also in other parts of the manuscript. It may be necessary to emphasise something, but please not too often! However, it is important to cite detailed information about the species characteristics, especially since the authors only detected the species via amplicon analysis and did not observe their genetic potential/portfolio.

The manuscript contains also many minor grammatical and wording mistakes that need to be corrected, see some examples below. Please go very carefully through the manuscript and correct accordingly!

Line 156: … as suggested per WHO?  Please place a correct sentence.

Line 234-237: here it would be nice to have a citation of the information source.

Line 237-240: Hydrogenophaga sp  are appearing in a lot of studies involved in sediment and surrounding water analyses (not just cold lakes); it seems to be associated with microalgal communities; but also here, it would be nice to back up the made statements with citations

Line 249: in the artic ? something seems to be missing

Lines 404-405: do not correspond, instead of not in concurrence

Comments on the Quality of English Language

The manuscript contains also many minor grammatical and wording mistakes that need to be corrected. Please go very carefully through the manuscript and correct accordingly!

Author Response

Comments and Suggestions for Authors

The manuscript “Analysis of culturable bacterial diversity of Pangong Tso lake by metagenomics approach” by Yadav et al. presents the analysis of the diversity of the culturable bacteria by applying a high throughput amplicon sequencing approach based on long read technology to determine the spectrum of bacterial diversity supported by axenic media.

  • The title is a bit misleading: metagenomics approach is not what the authors results provided, which was basically amplicon sequencing data based on the full length 16S rRNA gene (1.5Kbp covering regions V1-V9). Please correct accordingly!

Ans: The title has been modified as “Analysis of culturable bacterial diversity of Pangong Tso lake by 16S rRNA tag sequencing”

  • Line 4: The name of the corresponding author is missing; or at least “and” is misleading…

Ans: Done.

Materials and Methods:

  • 1. Sample collection: lines 67-68: The authors mentioned that the samples were transported to the laboratory within 48h after collection. This is quite a long time.. How were the samples stored in-between? There is a lot of research around confirming that even hours (of “unproper” storage) are enough to change the community composition, but definitely to affect the amount of detectable nutrients or carbon availability. It is highly crucial to keep the samples in the dark, and cold; in a “perfect world” directly frozen..., otherwise the community or nutrient composition might have changed...

Ans: The samples were stored in styrofoam boxes with ice packs and thus transported in dark and cold conditions to lab.

  • Another issue would be the composition of the used media for cultivation (please provide a table for the media components)

Ans: Supplementary table S5 for media composition included in revised version.

  • and were the plates cultivated in the dark or light; since many bacteria are capable of anoxygenic photosynthesis?

Ans: For cultivation, no specific light source was used in this study as our aim was to assess cultivable heterotrophic bacterial diversity using amplicon sequencing and did not target cultivation of phototrophs.

  • Was the analysis of the subsequent samples (especially PSD and PW) performed in replicates (preferably three independent DNA isolations)? 

Ans: Yes, community DNA from PSD and PW samples were isolated in triplicates and pooled for subsequent sequencing.

  • In general, the author should provide a separated view on the communities which were originally present in the sediment and the surrounding water (and present them in a Figure (or Table), maybe containing more information than just the phyla level) and then compare to the obtained communities via cultivation different media. Provide the data analysis in the main manuscript and not in the supplemental table.

Ans: Thank you for your suggestion. This information was already available in the submitted manuscript as genus level heat map comparison between the samples in terms of qualitative and quantitative data (Fig 3). But taking into account the reviewer’s suggestion and to provide more clarity to readers, in the revised manuscript we have incorporated comparative analysis of shared and unique taxa among different sample types i.e. sediment vs water, sediment vs sediment media samples and water vs water media samples in figure 4, Table S4 and line no 126-128 in materials and method and line nos 242-246 and 367-383 in results and discussion section of the revised manuscript.  

  • How many species could actually be retrieved by the culturing attempts? Itcould be a valuable information to get deeper insight into the bacterial diversity of Pangong Tso not just on phyla level and only for bacteria, but also cyanobacteria on class, family or genus levels. And maybe compare the most abundant taxa, especially with previous? Would there be (m)any correlations? 

Ans: Although culturing based analyses was performed for the same set of samples, this was not part of the manuscript because of limitations and scope of the present manuscript. However, in the revised manuscript we have presented this data as a comparative analyses between both approaches in the revised version (Fig S6). The necessary text is included in lines 398-400. As per our response to question 5, phototrophic bacteria such as Cyanobacteria were not targeted for cultivation.

In respect of reference to previous studies, pl refer to figure S3 of the original submitted manuscript for a phylogenetic tree and a bar graph that depicts qualitative and semi-quantitative information on taxa distribution (at genus level) with respect to present work and previous 3 reports of the lake ecosystem (Chaudhari et al., 2020, Bisht et al., 2018 and Rathour et al., 2020). The tree represents the most abundant taxa recovered in the present analysis and previously. Few abundant taxa i.e. Loktanella, Pseudomonas, Rheinheimera, Alishewanella, Flavobacterium, Belliella, Variovorax, and Comamonas were commonly identified in both previous reports and in present analysis. This fact has been mentioned in lines 265-328 of the revised manuscript.

  • There is no need to cite permanently the literature twice, since it interrupts the reading flow, e.g.  Line 151-152, but also in other parts of the manuscript. It may be necessary to emphasise something, but please not too often! However, it is important to cite detailed information about the species characteristics, especially since the authors only detected the species via amplicon analysis and did not observe their genetic potential/portfolio.

Ans: Corrected in line nos 133-134, 277, 310-311, 355, 364. For the comment “However, it is…..”, pl refer to line nos 254-342 that mentions detailed phenotypic characteristics as well as ecological features for abundant taxa detected in our analysis in results and discussion section.

  • The manuscript contains also many minor grammatical and wording mistakes that need to be corrected, see some examples below. Please go very carefully through the manuscript and correct accordingly!

Line 156: … as suggested per WHO?  Please place a correct sentence. 

   Ans: Grammatical corrections have been incorporated all through the manuscript to the best of our abilities and correction done for “WHO”; pl line no 163.

Line 234-237: here it would be nice to have a citation of the information source.

Ans: Pertinent citation added at line no 259.

  • Line 237-240: Hydrogenophaga spare appearing in a lot of studies involved in sediment and surrounding water analyses (not just cold lakes); it seems to be associated with microalgal communities; but also here, it would be nice to back up the made statements with citations

Ans: The lines have been reformatted in the revised version. We were only referring to non-occurrence of this taxa as a abundant genera in Pangong lake ecosystem. The relevant citations are incorporated (Line no 259-263). .

  • Line 249: in the artic ? something seems to be missing

Ans:Done line no 271-272.

  • Lines 404-405: do not correspond, instead of not in concurrence

 Ans: Done. line no 448.

Reviewer 4 Report

Comments and Suggestions for Authors

The authors have developed a microbial ecology study of an interesting environment: a high altitude, high salinity, high UV-receiving, extreme temperature-subjected lake.

The microbial ecology study was done during the month of September and over the water and sediments in the lake.

The culture-independent microbial identification used novel sequencing technology to provide reads of the full 16S rRNA gene of Bacteria, allowing for very fine taxonomic identification, which is both novel and scientifically powerful when considering these types of scenarios.

The bioinformatics approach for the sequencing sounds good to me and is much better than most works I review these days, so I want to congratulate the authors on that.

The description of the microbial ecology of the lake is deep, careful, methodic but interesting nonetheless.

I believe that the research has been well conducted and that the materials presented in the manuscript have good scientific soundness.

Thus, I want to congratulate the authors for their work.

On the other hand, I believe that the manuscript could benefit from some additions, which I reflect below:

1) I have not seen any particular statistical test that can discern characteristic microbes in water versus sediments, and also core phylotypes that appear in both. I think this could be a good addition that could even bring interesting things to discuss. For this reason I suggest the authors to proceed with differential abundance analysis (or similar tests) between the water and sediment samples to  check for these microbes.

2) Given the taxonomic accuracy that full sequencing of 16S rRNA gene provides, it might be possible to provide an accurate prediction of the metagenome present in the lake water and sediments. I suggest the authors to proceed with metagenome prediction using PICRUSt2 or some similar software. The predicted metagenome could also provide interesting points for discussion in the manuscript.

Author Response

Comments and Suggestions for Authors

The authors have developed a microbial ecology study of an interesting environment: a high altitude, high salinity, high UV-receiving, extreme temperature-subjected lake.

The microbial ecology study was done during the month of September and over the water and sediments in the lake.

The culture-independent microbial identification used novel sequencing technology to provide reads of the full 16S rRNA gene of Bacteria, allowing for very fine taxonomic identification, which is both novel and scientifically powerful when considering these types of scenarios.

The bioinformatics approach for the sequencing sounds good to me and is much better than most works I review these days, so I want to congratulate the authors on that.

The description of the microbial ecology of the lake is deep, careful, methodic but interesting nonetheless.

I believe that the research has been well conducted and that the materials presented in the manuscript have good scientific soundness.

Thus, I want to congratulate the authors for their work.

On the other hand, I believe that the manuscript could benefit from some additions, which I reflect below:

  • I have not seen any particular statistical test that can discern characteristic microbes in water versus sediments, and also core phylotypes that appear in both. I think this could be a good addition that could even bring interesting things to discuss. For this reason, I suggest the authors to proceed with differential abundance analysis (or similar tests) between the water and sediment samples to check for these microbes.

Ans: Thank you for your suggestion. For performing the statistical test, in the revised manuscript, unpaired t test with bonferroni correction by grouping all the sediment samples (total and media samples) and water samples (total and media samples) was completed (figure S5). The relevant portion has been incorporated in the manuscript text (line nos 378-383) in result discussion and materials and methods (line nos 128-131).

For the core phylotypes pl refer to our response for question 7 of reviewer 3.   

  • Given the taxonomic accuracy that full sequencing of 16S rRNA gene provides, it might be possible to provide an accurate prediction of the metagenome present in the lake water and sediments. I suggest the authors to proceed with metagenome prediction using PICRUSt2 or some similar software. The predicted metagenome could also provide interesting points for discussion in the manuscript.

Ans: As per the reviewer suggestion even though long read sequencing technology like nanopore are better for taxonomic analysis than short read-based technologies due to their higher informational content in the sequences yet majority of the main stream adopted classifiers rely on short read based algorithms by default that do not scale very well with long read data sets for functional prediction. The present samples were sequenced with old chemistry therefore we were unable to process the reads for predictive functional analysis even with newer tools like PICRUST2 or alternative contemporary softwares (Tax4Fun 2) new which have incorporated updated algorithms to predict the functional properties from long read sequencing technology. Further literature search to the best of our efforts did not yield any information on functional prediction from nanopore based amplicon data. Yet if the reviewer still feels this can be achieved, we are open and happy to learn and incorporate the required process for further enrichment of the data.

Round 2

Reviewer 1 Report

Comments and Suggestions for Authors

It's fine.

Author Response

Dear Reviewer,

Thanks for your constructive comments and feedback in improving the quality of our submission.

Regards

KMurthi

Reviewer 3 Report

Comments and Suggestions for Authors

Although the manuscript “Analysis of culturable bacterial diversity of Pangong Tso lake by metagenomics approach” by Yadav et al has improved considerably, but there is still room for improvement.

It would be good to get an overview of how many species or subspecies the authors were able to isolate or enrich using their cultivation strategy (perhaps expressed in per cent) compared to the initial samples (sediment and water). And possibly also list reasons with the help of the literature, e.g. vitamin auxotrophy, why (how many?) species could not be isolated?

Line 132-134, 150, 156-157, 272, 277 and elsewhere in the text: please use proper citations according to journals regulations (using numbers and not just names like for instance Chaudhari et al., 2020)  

Please italicize only family, genus, species, and variety or subspecies. There is no need to itacalize kingdom, phylum, class, order, and suborder of bacteria.

Figure 3: The legend mentions “distribution of genera level diversity in both direct metagenome samples and media samples” should be corrected (here and elsewhere) since the authors did not perform metagenome seq.

Figure 4: I don’t really understand which of the presented species are shared and which are unique among sediment and water samples. Please present the figure much clearer. Does the Fig 4 show all detected genera in the subsequent sample or just a selection; if yes based on which criteria?

Comments on the Quality of English Language

There are still many minor grammatical and wording mistakes that need to be corrected, see some examples below. Please go very carefully through the manuscript and correct accordingly!

E.g. Line 181: there is something wrong with the number “total of 2,30,129 reads were

Line 407: “In our culturable work…” what does that mean?

Author Response

Comments and Suggestions for Authors

Although the manuscript “Analysis of culturable bacterial diversity of Pangong Tso lake by metagenomics approach” by Yadav et al has improved considerably, but there is still room for improvement.

  • It would be good to get an overview of how many species or subspecies the authors were able to isolate or enrich using their cultivation strategy (perhaps expressed in per cent) compared to the initial samples (sediment and water). And possibly also list reasons with the help of the literature, e.g. vitamin auxotrophy, why (how many?) species could not be isolated?

Ans: As we had employed 16S rRNA tag sequencing for this study the maximum taxonomic resolution was limited at genus level. Thus, we are not able to distinguish the taxa at species or subspecies level in this present manuscript. Also, pl refer to our replies for your queries 5 and 8 in previous comment file, and to our reply to reviewer 2 wherein we have mentioned in detail the limitations of the present approach in terms of retrieving only limited diversity (i.e. only heterotrophs) because of the media formulations employed.

Further, in this study data pertaining to culturing-based analyses were excluded due to limitations and the scope of current manuscript (pl refer to our response to your query no 8 in the last comments file). However, as per suggestion in the revised manuscript, we had included culturable data to highlight a brief comparative analysis in the form of a new Figure S6 and line nos 416-418 for specific reference and lines 376-442 for comprehensive discussion related to previously retrieved cultured taxa from the same habitat and to our own culturable data of the same set of samples. But since this work is quite detailed and in the process of being written as part of a separate manuscript therefore, we are not in a position to elaborate further in this respect. 

  • Line 132-134, 150, 156-157, 272, 277 and elsewhere in the text: please use proper citations according to journals regulations (using numbers and not just names like for instance Chaudhari et al., 2020) 

Ans:  Done, pl see line nos 133, 134, 151, 157, 158, 163, 269, 274, 307-308, 332, 361, and 455-456 in the revised version.

  • Please italicize only family, genus, species, and variety or subspecies. There is no need to itacalize kingdom, phylum, class, order, and suborder of bacteria.

Ans: Thank you for your corrections we have promptly corrected the mistakes all through the manuscript.

  • Figure 3: The legend mentions “distribution of genera level diversity in both direct metagenome samples and media samples” should be corrected (here and elsewhere) since the authors did not perform metagenome seq.

Ans: The correction has been made in line no 238, 347, 453, and 454.

  • Figure 4: I don’t really understand which of the presented species are shared and which are unique among sediment and water samples. Please present the figure much clearer. Does the Fig 4 show all detected genera in the subsequent sample or just a selection; if yes based on which criteria?

Ans: We have modified figure 4 and its legend for better representation of the figure. In figure 4: A, C and E represent unique and shared taxa (at genus level) between the samples whereas B, D and F represent the identities of the shared taxa between the samples. As mentioned in line no 128 the taxa <0.1% were excluded for this analysis and taxa greater than 0.1% was used for shared and unique taxa analysis.  

Comments on the Quality of English Language

6)  There are still many minor grammatical and wording mistakes that need to be corrected, see some examples below. Please go very carefully through the manuscript and correct accordingly!

E.g. Line 181: there is something wrong with the number “total of 2,30,129 reads were…

Ans: Done. As far as possible we have tried to remove grammatical errors in the revised version.

Line 407: “In our culturable work…” what does that mean?

Ans: The sentence has been re framed in line no. 400. We intended to mention the details of the cultured taxa that were isolated and purified from the samples. Also refer to our response to query 1 above.